# LGBTQ+ Youth’s Identity Development in the Context of Peer Victimization: A Mixed Methods Investigation

**DOI:** 10.3390/ijerph19073921

**Published:** 2022-03-25

**Authors:** Sarah Kiperman, Hannah L. Schacter, Margaret Judge, Gabriel DeLong

**Affiliations:** 1Theoretical and Behavioral Foundations Division, Wayne State University, Detroit, MI 48202, USA; gdelong@wayne.edu; 2Department of Psychology, Wayne State University, Detroit, MI 48202, USA; hannah.schacter@wayne.edu; 3Merrill Palmer Skillman Institute for Child and Family Development, Detroit, MI 48202, USA; 4Department of Psychology, Cedarville University, Cedarville, OH 45314, USA; margaretjudge@cedarville.edu

**Keywords:** LGBTQ+ youth, peer victimization, identity development, social support, outness, mixed methods

## Abstract

Research rarely explores LGBTQ+ youth bullying in the context of culture-specific outcomes (e.g., LGBTQ+ identity development) and what can mitigate the impact of peer stressors. This study used a concurrent mixed methods design to explore how experiences of peer victimization predicted LGBTQ+ youth’s identity development (i.e., stigma sensitivity, concealment motivation, and difficult process) and whether social support and outness served as protective, moderating factors. The mixed methods approach provides a culture-specific context via qualitative inquiry to inform whether the quantitative findings align with how youth qualitatively discuss their experience of peer victimization, negative outcomes, and social support. Our sample consisted of 349 LGBTQ+ youth 14–17 years old who completed a survey (quantitative sample) and a subset of 39 LGBTQ+ youth who completed a semi-structured interview (qualitative sample). Our quantitative findings indicated that greater overall peer victimization was positively related to LGBIS-revised subscales of stigma sensitivity, concealment motivation, and difficult process, where both outness and social support moderated such relations. Qualitatively, victimized youth also reported stigma sensitivity and concealment motivation while also endorsing how being out and having a support system played a role in their experience of being victimized. These qualitative findings align with our quantitative findings that classmate support mitigated the effects of peer victimization on the difficulty of coming out. Implications for practitioners and researchers are provided.

## 1. Introduction

Although a major milestone of adolescence is establishing autonomy [1], Lesbian, Gay, Bisexual, Transgender, and Questioning/Queer+ (LGBTQ+) youth identity development models acknowledge the added unique challenges of making sense of one’s diverse sexual orientation or gender identity as well as negotiating their outness to themselves and others [2,3]. The “+” in the above LGBTQ+ acronym aims to include sexual and gender minority community members not exclusively named in the acronym. Researchers [4] further conceptualized LGBTQ+ identity development by developing the Lesbian, Gay Bisexual Identity Scale (LGBIS), which documents critical perceptions that LGBTQ+ individuals may encounter as an LGB community member. These factors and their definitions include: (1) identity dissatisfaction, how satisfied one is with being LGB, (2) identity uncertainty, how confused or stable one is in seeing themselves as LGB, (3) stigma sensitivity, how anxious LGB people are around being rejected based on being LGB, (4) identity centrality, the focus of sexuality as a central part of one’s life, (5) difficult process, how difficult coming out and being LGB is for people, (6) concealment motivation, how motivated one is to conceal or hide their LGB identity rather than being out, and (7) identity superiority, seeing one’s LGB identity as superior to others. 

It is important to specifically understand that LGBTQ+ identity development is susceptible to impact from minority stressors. Researchers [5] discuss that people who are susceptible to discrimination (with marginalized identities) experience additional stressors above and beyond people without marginalized identities and, as a result, tend to have poorer health outcomes. This is especially relevant for LGBTQ+ youth as they experience stressors such as peer victimization [6] at greater rates than their heterosexual, cisgender peers. Research has, in turn, documented how encountering these contextual stressors relates to LGBTQ+ youth demonstrating greater rates of depression [7], suicidal ideation [8], and truancy [9], compared to their heterosexual, cisgender peers. Related to their identity development, research has identified how bullying can impact sexual minority youths’ self-esteem and how internalized homophobia can partially function as a mediator in this context [10]. Given these findings and the potential impact of additional stress on LGBTQ+ youth, we should seek to further understand LGBTQ+ ‘culture-specific identity development’ in the context of adverse experiences such as peer victimization. 

### 1.1. Peer Victimization

Peer victimization refers to the physical, verbal, or psychological abuse of victims by peer perpetrators who intend to cause them harm [11,12]. Youth may use a variety of tactics to hold power over others, such as popularity, physical strength, and embarrassing information [11,12]. Peer victimization can be physical (intentionally causing bodily harm to others or destroying their possessions; this can include hitting, kicking, tripping, taking/breaking possessions), verbal (using written or spoken words to hurt others, such as threatening, calling names, or teasing), social/relational (hurting others in their social standing via intentional exclusion, telling others to not be friends with someone, starting/spreading rumors), and cyberbullying (intentionally causing harm or embarrassment on social media platforms, instant messaging, or gaming systems) [11,12]. LGBTQ+ youth are exposed to increased prevalence rates of bullying compared to their heterosexual, cisgender peers [6]. Furthermore, LGBTQ+ youth who experienced school-based victimization demonstrated a significantly reduced sense of high school belonging, increased endorsement of depressive symptoms, and significantly increased feelings of general psychological distress in young adulthood [13]. Additionally, the school climate survey identified percentage rates at which LGBTQ+ youth experienced cyberbullying (44.9%), verbal harassment (68.7%), physical harassment (25.7%), physical assault (11%), property damage (35.7), and relational bullying (90.1%, [14,15]). Bullying increases the chances of an LGBTQ+ youth attempting suicide by 300% relative to their LGBTQ+ peers who are not bullied [10].

Researchers underscored the importance of stigma sensitivity, concealment motivation, and difficult processes in the context of the LGBTQ+ youth identity development experience and victimization. Research [10] discussed stigma sensitivity as internalized homophobia, where youth who experience more bullying perceive greater internalized homophobia and lower self-esteem. Concealment motivation was described as visibility management, where youth make careful decisions about who they are out to in various settings and circumstances [16,17]. A qualitative study identified how LGBTQ+ youth might not come out to maintain the privileged façade of heterosexual identity, thus limiting their exposure to LGBTQ+ specific victimization and discrimination [17]. Research also discussed the difficult process of coming out for LGBTQ+ youth as many become exposed to LGBTQ+ specific bullying, discrimination, and additional minority stress [5,18]. Thus, while we know some information about the role that identity development plays in the lives of LGBTQ+ youth, how peer victimization predicts LGBTQ+ identity development factors, as well as how it may be moderated by additional factors (e.g., social support and outness), is relatively unexplored.

### 1.2. Combatting Peer Victimization: Social Support as a Buffer

To combat the undue stress of peer victimization, many LGBTQ+ youth report relying on their social support network. Social support is comprised of positive influences from those who offer LGBTQ+ youth compassion, care, information, or assistance [18]. Although social support can be beneficial for many LGBTQ+ youths in the absence of other stressors (i.e., main effect), social support can also serve a stress-buffering function, where youth lean on their support network as a metaphorical shield to cope with stressors in their lives [19].

Several studies identified the effectiveness of social buffering against adverse experiences for LGBTQ+ youth. Peer support was identified as a moderating buffer against negative attitudes from family members toward LGB youth ages 17–27; and it also buffered their experience of anxiety, depression, and perceived victimization [20]. Despite these findings, additional research found that family but not peer support in childhood could moderate the effects of homophobic victimization on anxiety and physical pain in emerging adulthood [21]. A mediation model for transgender adults identified that perceiving increased harassment, rejection, and discrimination predicted suicidal ideation most strongly when participants perceived having low social support from a significant other in comparison to moderate or high support [22]. Thus, having low or no social support related to their increased mental health concerns.

Several studies focused on the role of specific people in LGBTQ+ youth’s lives and how they may buffer against bullying exposure [18,23]. One study explored how different combinations of support sources for LGBT youth ages 16–20 predicted mental health outcomes when controlling for one’s lifetime exposure to LGBT-related victimization [23]. In designating different combinations of support sources, this study [23] designated three clusters: low support (feeling little to no support from family, peers, and significant other), non-family support (feeling little to no support from family, but moderate support from peers and significant other), and high support (feeling adequately supported by family, peers, and significant other). Findings indicated youth who endorsed high support reported feeling significantly less hopelessness, loneliness, depression, anxiety, somatization, suicidality, and global severity compared to youth who endorsed having low support. The findings associated with having low support were consistent for youth in the non-family support cluster; however, there was no difference in the hopelessness and anxiety levels for youth in the non-family support cluster. The School Climate Survey [18] explored LGBTQ+ youth perceptions of support in their schools. They reported LGBTQ+ youth found emotional support (73.8%) and socializing opportunities (87.5%) from their gay–straight alliances. Fortunately, 97.7% of LGBTQ+ youth respondents identified at least one school staff member supportive of LGBTQ+ students, and 66.3% identified six or more supportive school staff. Additionally, 48.4% of respondents shared they could identify at least one LGBTQ+ staff person who was out at their school.

### 1.3. Peer Victimization: Outness as a Moderator

Outness for LGBTQ youth refers to “the extent to which one’s LGBTQ+ identity is known to others” [24]. Findings identified a significant positive relationship between one’s outness to others and their exposure to victimization [15,24,25]. A possible explanation for this is that LGBTQ+ youth tend to be targeted in bullying at greater rates [6], and being out could, in turn, make youth more susceptible to victimization. Conversely, while LGBTQ+ youth who endorsed being out to peers and staff at school experienced more in-school victimization, youth also endorsed that being out related to them having higher self-esteem, decreased depression, and missing fewer days of school [15]. These findings indicate that while being out is related to increased victimization, there are also social-emotional benefits such as perceived relief around being one’s authentic self and sharing that with others. While we have some insight around the role of outness in LGBTQ+ youth’s victimization experiences, few studies explore how outness may moderate the effects of peer victimization on LGBTQ+ youth identity development.

### 1.4. Rationale for Mixed Methods and Transformative Framework

Research thus far on LGBTQ+ youths’ experiences of bullying predominantly uses quantitative methods. While quantitative methods can convey prevalence rates, significant relationships, and generalizable findings [26], relying on these findings often lacks the context of a culture-specific perspective found in qualitative research [27] and relies on measures procured by researchers rather than participants. Using a mixed methods approach permits researchers to integrate independently accrued quantitative and qualitative methods to inform generalizable and contextually specific perspectives, which can yield a more valid representation of participant perspectives [28]. Specifically, via a convergent mixed methods approach, researchers gather qualitative and quantitative data simultaneously and independently with the intent of identifying where findings converge, diverge, or expand on each other. Using a mixed methods design in the context of LGBTQ+ youth peer victimization would allow researchers to learn how the measures we use to convey experiences of peer victimization, social support buffering, outness, and outcomes reflect how LGBTQ+ youth discuss these experiences.

When using mixed methods, researchers present their guiding framework [28]. This study is guided by the transformative paradigm [29] and the community-based participatory research (CBPR) approach [30]. The transformative framework presents a socially just framework that values conveying culture responsiveness as lack of representation is often impacted by societal power imbalances. The framework also values having research that is based on trusting relationships among research and the represented culture, as well as conducting mixed methods research aimed at achieving social change [29]. We also engage the CBPR approach as it aims to foster an equitable partnership among community members, organizational representatives, and researchers, where all partners contribute to decision making throughout research- and action-based processes [30].

### 1.5. Current Study

While LGBTQ+ youth encounter additional minority stress, such as victimization, compared to their heterosexual, cisgender peers, its contextual impact on LGBTQ+ identity development is relatively unexplored. Moreover, although we know that LGBTQ+ youth rely on social support to buffer against adverse experiences [31], it remains unclear how social support and outness function as moderators in buffering LGBTQ+ youth’s identity development. Furthermore, studies to date have not used a mixed methods approach to understand the LGBTQ+ youth experience of bullying. By integrating qualitative and quantitative findings via mixed methods analysis, we can obtain a more culture-specific understanding of this phenomenon. Thus, this paper uses a mixed methods approach to examine the relationships between these constructs and explore how youth discuss their experiences using ethnographic interview methods. Our investigation asks the following research questions (RQ):

QUAN: (1) How are experiences of peer victimization related to LGBTQ+ youth identity development? and (2) How does outness and support from peers and friends moderate the relationship between peer victimization and LGBTQ+ youth identity development? We expect that more exposure to peer victimization will lead to increased stigma sensitivity, perceived difficult processes, and increased concealment motivation in alignment with previous findings [10,15,16]. We further hypothesize that social support and outness will buffer against or moderate the relationship between peer victimization and LGBTQ+ youth identity development outcomes. These hypotheses are informed by previous findings that already point to the buffering potential of social support [20,21,22,23] and from the higher prevalence rates of LGBTQ+ specific bullying that may target out youth at greater rates [24,25].

QUAL: (1) How do LGBTQ+ youth experience peer victimization? (2) How do LGBTQ+ youth experience their identity development in the context of peer victimization? (3) How do LGBTQ+ youth experience their outness in the context of peer victimization? and (4) How do LGBTQ+ youth experience social support in the context of peer victimization?

While these questions are exploratory, we hypothesize that for RQ1, LGBTQ+ youth participants will discuss experiences of verbal, physical, and relational victimization as previously identified in the literature [14]. We hypothesize for RQ2 that LGBTQ+ youth who experience peer victimization will discuss a negative impact on their identity with possible themes of internalized homophobia [10]. We hypothesize for RQ3 that LGBTQ+ youth who experience peer victimization will unfavorably reflect on their decision to be out with their peers, in alignment with [16] findings. We also hypothesize for RQ4 that LGBTQ+ youth will express buffering properties from peers when exposed to bullying [21,22,23].

MIXED METHODS: (1) How do qualitative findings converge, diverge, or expand on quantitative findings in conveying LGBTQ+ youth’s experience of peer victimization and how do they relate to their identity development when moderated by social support and outness, in determining how to foster healthy identity development for LGBTQ+ youth? We hypothesize that findings will mostly converge, where culture-specific context will be offered from qualitative data; however, we expect to find divergent findings related to how one’s outness moderates the relationship of peer victimization with identity development. The literature currently depicts being out as making one more susceptible to victimization due to higher rates of targeted victimization towards LGBTQ+ youth [10,15,16]; however, we expect to find being out can facilitate youth being their authentic selves, which may present with buffering features similar to social support [20,21].

## 2. Materials and Methods

Guided by the transformative paradigm [29] and the CBPR approach [30], we used a concurrent mixed methods design (Figure 1, [28]). We equally relied on qualitative and quantitative data that we gathered simultaneously, where the quantitative data informed (1) our deductive understanding of how victimization related to their LGBTQ+ identity development (e.g., internalized stigma and perceived difficulty of the process of coming out to oneself) and (2) how social support and being out moderates or buffers against one’s negative perceptions of their LGBTQ+ identity development when experiencing victimization. The qualitative data informed our inductive understanding of (1) the victimization LGBTQ+ youth report experiencing, (2) how they discuss their identity development when they report experiencing victimization, and (3) how they discuss social support and outness as buffering against their experiences of victimization. The study’s qualitative and quantitative data collection and analysis were conducted concurrently with the goal of engaging multiple perspectives to produce a nuanced explanation of LGBTQ+ youths’ experiences of victimization, the impact it may have on their identity development, and how social support and outness may buffer against negative perceptions of identity development. A second goal was to generate a culture-specific understanding of LGBTQ+ youths’ victimization experience using a mixed methods approach so we can identify times we could intervene and mitigate negative outcomes to contribute to social change [29].

### 2.1. Participants

Sample recommendations for qualitative semi-structured individual interviews call for 15–20 participants per sample [32]. Researchers engaged *N* = 39 participants in qualitative interviews and surveys, with an additional *n* = 310 participants also completing the survey, yielding a total quantitative sample of *N* = 349. These sample sizes meet qualitative saturation standards [32] and meet quantitative power expectations for analyses [33]. Eligible participants needed to be 14–17 years old, identify as LGBTQ+, be in high school, and have an email address to receive study-related emails.

In our quantitative sample of *N* = 349, youth participated via waived (*n* = 248) and parent (*n* = 101) consent; there were *n* = 70 9th graders, *n* = 101 10th graders, *n* = 86 11th graders, and *n* = 79 12th graders. Youth self-reported being lower (*n* = 33), working, (*n* = 81), middle (*n* = 168), upper-middle (*n* = 61), and upper class (*n* = 3); living in urban (*n* = 90), rural (*n* = 69), and suburban (*n* = 186) settings; and received school services in home (*n* = 17), private (*n* = 29), public (*n* = 244), charter (*n* = 10), alternative (*n* = 10), online (*n* = 32), and GED (*n* = 3) settings. In our qualitative sample of *N* = 39, youth participated via waived (*n* = 17) and parent (*n* = 22) consent; there were *n* = 10 9th graders, *n* = 9 10th graders, *n* = 8 11th graders, and *n* = 12 12th graders. Youth self-reported being lower (*n* = 3), working, (*n* = 13), middle (*n* = 16), upper-middle (*n* = 6), and upper class (*n* = 1); living in urban (*n* = 13), rural (*n* = 9), and suburban (*n* = 17) settings; and received school services in home (*n* = 1), private (*n* = 3), public (*n* = 31), charter (*n* = 1), alternative (*n* = 1), online (*n* = 2), and GED (*n* = 0) settings. See Table 1 for additional participant demographic information (e.g., age, sexual orientation, and gender identity).

Researchers first recruited LGBTQ+ youth to complete both the interview and survey. Researchers used purposive sampling, where participants were recruited at participating locations [34]. Researchers contacted more than 32 LGBTQ+ affirming community-based organizations (CBOs) and schools, a majority of which were located in a major metropolitan city in the Southeast United States and of which ultimately four community organizations and two schools participated. To recruit, youth attended recruitment presentations hosted by the partnering organizations where researchers presented a scripted overview of the study. They viewed flyers on the study in the community centers and on social media, talked to researchers at social events they attended through partnering organizations (e.g., PRIDE, GSA Youth Summit, and school assemblies), and shared study information with others via word of mouth also known as snowball sampling [35]. Interested youth contacted researchers via the contact information provided at recruitment events, where researchers then screened youth to determine their eligibility and scheduled their interview/survey participation.

Additional youth were recruited for the quantitative survey sample with purposive sampling [34] via Instagram social media posts. Posts were purposive via the hashtags we used to recruit (e.g., #LGBT #LGBTQ #LGBTPRIDE #LGBTYOUTH #GSA #GAYBOY #QUEER #LESBIAN).

### 2.2. Procedures

Research procedures were reviewed and approved by the university’s institutional review board. Youth who were interested in participating contacted researchers in person or via the phone number or email address listed on recruitment materials. Youth shared whether they could secure parent/guardian consent or if they wanted to use waived consent to avoid harm or outing [36]. Youth who opted for parent/guardian consent received a parent/guardian consent form via email. Those who opted for waived consent reviewed their assent with a youth advocate. A youth advocate was a designated adult at each CBO that partnered with this study who answered questions, reviewed the research with youth, and did not have a direct investment in the study’s outcomes [36]. Returning consent forms or reviewing assent with a youth advocate occurred prior to their participation.

For participants completing qualitative interviews and surveys, one of four research assistants trained in qualitative interviewing interviewed participants. Researchers conducted interviews at the participant’s CBOs and schools, the researcher’s university, or if out of state, via Skype with a login created by the study. Interviews were audio-recorded by the research assistants. Participants received a $15 stipend for their time. Research assistants transcribed interviews verbatim and then transferred the transcripts for coding to MaxQDA qualitative data software for analysis [37].

For the remaining quantitative surveys, youth responded to an eligibility survey they accessed through an LGBTQ youth-targeted Instagram ad [34]. If youth were eligible, researchers sent them contact information for a youth advocate to talk through the assent form or a parental consent form for them to return, as well as a link to the survey. Youth assent was nested within the survey, and youth who could access parent consent returned signed forms via email. Youth received a $5 Starbucks e-gift card for their time via email.

### 2.3. Data Analysis

This study used a concurrent mixed methods approach. This indicates that qualitative and quantitative data were collected independently by qualified investigators in each respective skillset and were then merged and synthesized as a team, guided by a researcher with significant mixed methods experience. The following sections on gathered data discuss qualitative and quantitative data separately. Figure 1 present the mixed methods process of independently engaging in each data gathering method, show when and how the researchers merged the methods, and also include a notation that conveys our equal value on concurrent qualitative and quantitative methods in this study, and our intent to mix findings (=).

### 2.4. Qualitative Methods

#### 2.4.1. Qualitative Instruments

Participants completed a 20 min survey, which was a part of a larger study, and then a semi-structured interview that was designed to last for one hour (M = 54 min; ranging from 31 to 83 min in length). The interview was designed to learn how the LGBTQ+ participants perceived their social support and nonsupport experiences (interview protocol and questions are available from the senior author by request). Researchers asked the youth to “tell me about your experiences of when you felt supported?” and “tell me about your experiences of when you felt not supported”. To yield rich, specific content, researchers asked participants prompts such as “tell me more,” naming keywords, and “what does that look like”.

#### 2.4.2. Qualitative Data Analysis

##### Qualitative Coding

Two research assistants and the first author conducted coding. Coding occurred as a recursive, inductive-deductive process where researchers inductively derived themes that aligned with participants’ perceptions from their interviews and deductively incorporated existing research to ground deductive themes in literature when relevant [38]. Researchers independently open coded (developed themes, their definitions) transcripts, then compared their open codes to inform a codebook that synthesized their open codes [39]. The researchers reached saturation (where no new ideas were generated) after the 20th transcript, after finding no new open codes since transcript 15 when they concluded their open coding procedures [38,39,40]. The researchers deductively rounded out the codebook by situating the open codes within social support and victimization frameworks posited by the literature [41,42].

Researchers then applied the codebook codes (via coding independently and then comparing their codes together) to transcripts using MaxQDA qualitative software [43] to determine the codebook’s reliability and where changes in the codebook needed to occur to yield a reliable, finalized codebook [44]. The codebook included the code name, a brief definition (to summarize the code), a full definition (to address nuances), and examples of when to apply or not apply the code [44,45]. When comparing codes, researchers reached a consensus and amended the codebook to prevent the discrepancy from reoccurring [46]. Researchers designated coding blocks or the start and endpoints of when to designate a code [47,48]. Researchers completed this process until they reached an 80% interrater agreement (a reliability method that informs the percentage coders agreed in their coding patterns, regarded as 80%), which occurred at transcript 10 where they attained an interrater agreement of 91.22%.

In alignment with practice recommendations [46], which called for researchers to return to transcript one and began the coding process again with their finalized codebook. By transcript three, researchers reached 90% interrater agreement (IRA) and began coding independently. The researchers used coder drift practices to ensure maintained reliability. Coder drift refers to coders’ tendency to change their interpretations of coding schemes as they code independently. Researchers tracked coder drift by calculating IRA for 10% of each individually coded interview [38]. If researchers achieved less than 80% coder drift, they had to review the entire transcript. Researchers attained an overall IRA of 91.68% and an overall coder drift IRA of 92.23%. All reviewed discrepancies were discussed by researchers until 100% consensus was reached [46].

##### Reviewing Codes for Current Analysis

The researchers identified a select number of codes that applied to the current analysis from the original codebook, particularly those that related to bullying, social support, and one’s self-concept in the context of their outness and their LGBTQ+ identity. MaxQDA permits researchers to identify codes as they occur and codes that occur simultaneously. In our analysis, we both reviewed independent codes (e.g., social support) and codes in the context of LGBTQ+ youth’s experience of being victimized (e.g., times LGBTQ+ youth participants discussed social support and victimization).

##### Trustworthiness

The current study incorporated multiple procedures to address the trustworthiness of the data (i.e., reliability, validity, and objectivity; [47]). The researchers enacted trustworthiness procedures already discussed, such as interrater agreement methods [46] and codebook development [45,46]. Researchers used additional trustworthy methods such as training self-checks (accounting for one’s worldview and biases; [29], audit trails (notetaking on the method’s process), and peer debriefings (discussions with an impartial peer to guide critical thinking and conceptualization of analysis; [47]).

The first author trained one other research assistant to conduct interviews by reviewing the protocol and conducting mock interviews [49,50]. Researchers practiced various interview procedures (e.g., body posture, person-centered responses such as summaries and paraphrasing) to reduce power differentials and build rapport [51,52].

Researchers used self-awareness check-ins to acknowledge their preconceptions that may have influenced their methods [29]. Research team member 1 (RTM1) is a white, straight female school psychology faculty who acknowledged her motive to further a social justice agenda; RTM2 is a white, straight female developmental psychology faculty with experience in quantitative methods and an ongoing commitment to conducting socially just research. RTM3 is a white, straight female undergraduate psychology student who was learning about mixed methods and qualitative research, and RTM4 is a multiracial (black and white), gay male graduate counseling psychology student who acknowledged his commitment to advocating for LGBTQ+ youth. Maintaining awareness of our worldviews helped account for biases throughout the study [29].

Audit trails [47] included detailed documentation of data collection, coding, data analysis, summaries of participant qualities (e.g., one participant stated they had autism), and takeaways from each interview to recall participants. Researchers also documented each version of the codebook as it was developed, noting revisions. The researchers engaged in each step of the study-maintained audit trails.

Peer (or mentor) debriefing was accomplished by research advisors to review the research process and data [47]. The first author sought support from her mentors and also served as a mentor to RTM3 and RTM4. Mentorship often involved challenging researchers’ assumptions and interpretations and providing feedback during study development, data collection, data analysis, interpretation, and dissemination. The researchers peer debriefed following each conducted interview and during codebook development and implementation to review coding choices.

### 2.5. Quantitative Methodology

#### 2.5.1. Measures

##### LGBIS

The Lesbian, Gay, and Bisexual Identity Scale (LGBIS) is a 27-item self-report measure that aims to assess LGB identity based on clinical and theoretical literature [4]. To align with the literature, we explored three domains within the scale: concealment motivation (i.e., I prefer to keep my same-sex romantic relationships rather private; my sexual orientation is a very personal and private matter)*,* internalized stigma sensitivity (i.e., if it were possible‚ I would choose to be straight; I believe it is unfair that I am attracted to people of the same sex), and difficult process (i.e., admitting to myself that I am an LGB person has been a very painful process; I have felt comfortable with my sexual identity just about from the start). Each dimension consists of an average of at least three items. Each statement is measured on a six-point Likert scale where one represents “Disagree Strongly”, and six reflects “Agree Strongly”. The LGBIS was normed on 1004 self-identified lesbian and gay adults aged 18 to 69. Participants were predominantly white, middle class, had at least a bachelor’s degree, and were from the U.S. West Coast. A six-week test–retest correlation indicated that the LGBIS subscales scores had adequate reliability and validity for research.

##### Peer Victimization

Past-year peer victimization was assessed using the Multidimensional Peer Victimization Scale, which asks participants to report their frequency of experiencing 16 types of peer victimization over the past year [53]. The 16 items are responded to on a 3-point scale ranging from 0 (Not at all) to 2 (More than once). The items capture different forms of victimization, including physical (e.g., “Punched me”), verbal (e.g., “Called me names”), social manipulation (e.g., “Made other people not talk to me”), and attacks on property (e.g., “Tried to break something of mine”). The 16 items were summed to yield an overall score of past-year peer victimization, where higher scores indicate more frequent victimization (α = 0.93).

##### Child and Adolescent Scale of Social Support-Revised (CASSS-R)

The CASSS-R is a 60-item Likert rating scale that assesses K-12 youths’ perceived social support from five sources: parents, teachers, classmates, close friends, and people in their school; and four content types of support: informational, instrumental, appraisal, and emotional [54]. Likert ratings are retrieved for two types of answers: the support’s frequency and importance to the participant. The frequency ratings were assessed via a 6-point Likert scale with answers ranging from “never” to “always”, while the importance of the support was assessed via a 3-point Likert scale, with answers ranging from “not important” to “very important”. The total support value was derived by summing all support frequency items. An overall importance value can result from summing all importance items but was not used in this study.

The CASSS-R is composed of five network subscales, which correspond to each network source of assessed support (parents, teachers, classmates, close friends, and people in their school). Each network subscale contains a total of 12 questions, where three questions within each subscale correspond to the content types: emotional, tangible/instrumental, informational, and appraisal. Network subscales are assessed by summing the total frequency ratings for each network source. Subscales for content types can also be derived by summing the three questions pertaining to the same support type from each source (a total of 15 questions).

Reliability analysis from a former study yielded Cronbach’s alpha coefficients of above 0.97 for the total frequency and total importance [54]. The Cronbach’s alpha coefficient for the subscales ranged from 0.92 to 0.96 and 0.89 to 0.95 for the frequency and importance subscales, respectively. Researchers found test–retest reliability evidence without the School Subscale for the frequency scores with coefficients of 0.75 for overall support score and ranges from 0.58 to 0.74 for subscales [54]. The CASSS-R was demonstrated to be sufficiently reliable with other measures of social support, such as the Social Support Scale for Children [55] and the Social Support Appraisals Scale [56].

##### Outness Inventory

The Outness Inventory (OI) is an 11-item self-report measure that describes the degree to which an individual has revealed their sexual and gender identity to others. The OI has 4 domains: outness to family, world, religion and overall outness [57]. The overall outness domain is comprised of the average scores of outness to family, world, and religion. Each statement is measured on a seven-point Likert scale where one indicates “definitely does NOT know about my sexual orientation status”, and seven represents “definitely knows about your sexual orientation status, and it is openly talked about”. The inventory also has the option of “not applicable”, which is represented with a zero. The OI was normed on a subsample of 411 lesbian and gay adults aged 18 to 69 and was found to show good support for reliability and validity. Researchers elected to use this measure with a youth sample, as items presented as relevant based on previous qualitative interviews with youth and given its sound quality. 

#### 2.5.2. Data Analysis

Data analysis proceeded in several steps. First, we calculated bivariate correlations and descriptive statistics for the main variables of interest. Second, to assess the effects of peer victimization on LGBTQ+ adolescents’ identity development, a series of linear multiple regressions were performed using SPSS to evaluate whether the frequency of peer victimization was related to four different aspects of LGBTQ+ adolescent identity development: stigma sensitivity, difficult process, concealment motivation, and identity dissatisfaction. Lastly, to evaluate potential protective factors, we examined adolescents’ outness to straight friends and peer social support as moderators of links between peer victimization and identity outcomes using the SPSS PROCESS macro. Significant interactions were probed by estimating simple slopes at low (−1 SD) and high (+1 SD) levels of the moderator.

All regression and moderation analyses controlled for participants’ age, gender identity, race/ethnicity, and geographic location. Gender identity was represented by three dummy codes (male; female; transgender), while students identifying as “other” (e.g., non-binary; non-conforming), the largest group in the sample, served as the reference group. Race/ethnicity was represented by six dummy codes (black; Asian; Latinx; Native American; multiracial; other), where students identifying as white, the largest group in the sample, served as the reference group. Geographic location was represented by two dummy codes (urban; rural), where students in the largest group in the sample, those who live in suburban locations, served as the reference group.

## 3. Results

In our results section, we answer our research question by presenting the quantitative findings first, followed by qualitative findings. We then triangulate qualitative and quantitative findings in a joint display to demonstrate how they converge or diverge [46,58].

### 3.1. Quantitative Results

#### 3.1.1. Descriptive Statistics and Bivariate Correlations

Descriptive statistics and bivariate correlations for the main study variables are presented in Table 2.

#### 3.1.2. Associations between Peer Victimization and LGBTQ+ Identity Outcomes

In the next set of models, we examined associations between peer victimization and four different LGBTQ+ identity outcomes: stigma sensitivity, (coming out as) difficult process, concealment motivation, and identity dissatisfaction. All models again controlled for participant gender identity, ethnicity, and geographic location. The results of these models are presented in Table 3. Over and above the effects of covariates, there were significant positive effects of peer victimization on stigma sensitivity (*b* = 0.08, *p* < 0.001) and difficult process (*b* = 0.06, *p* = 0.016). That is, LGBTQ+ youth who experienced more peer victimization were more likely to worry about others judging them for their sexual orientation or gender identity and more likely to experience coming out as a difficult process. However, there were nonsignificant effects of peer victimization on concealment motivation (*b* = 0.02, *p* = 0.492) and identity dissatisfaction (*b* = −0.00, *p* = 0.962).

#### 3.1.3. Moderating Effects of Outness and Social Support

To determine whether outness (to old and new straight friends) may moderate the effect of peer victimization on LGBTQ+ identity outcomes, the same four models were re-estimated with two interaction terms added: peer victimization X outness to old straight friends and peer victimization X outness to new straight friends. For the model predicting stigma sensitivity, there were nonsignificant peer victimization X outness to old straight friends (*b* = 0.00, *p* = 0.663) and peer victimization X outness to new straight friends (*b* = −0.00, *p* = 0.663) interactions. For the model predicting difficult process, there were also nonsignificant peer victimization X outness to old straight friends (*b* = −0.01, *p* = 0.361) and peer victimization X outness to new straight friends (*b* = −0.01, *p* = 0.209) interactions. That is, youth who experienced greater peer victimization also reported greater stigma sensitivity and a more difficult coming out process, regardless of how out they were to old or new straight friends. For the model predicting concealment motivation, there was a significant peer victimization X outness to new straight friends interaction (*b* = −0.02, *p* = 0.008), such that greater peer victimization predicted greater concealment motivation for youth who reported lower levels of outness to new straight friends (*b* = 0.07, *p* = 0.012) but not for those who reported higher levels of outness to new straight friends (*b* = −0.03, *p* = 0.229). In other words, peer victimized youth were less likely to feel motivated to conceal their LGBTQ+ identity if they were more out to their new straight friends. Finally, for the model predicting identity dissatisfaction, there was nonsignificant peer victimization X outness to old straight friends (*b* = −0.00, *p* = 0.978) and peer victimization X outness to new straight friends (*b* = 0.01, *p* = 0.574) interactions, indicating that peer victimization was unrelated to identity dissatisfaction regardless of youth outness to old or new straight friends.

To determine whether peer social support (i.e., from classmates or close friends) may function as a protective factor that buffers the effect of peer victimization on LGBTQ+ identity outcomes, the same four models were re-estimated with two different interaction terms added: peer victimization X classmate social support and peer victimization X close friend social support. For the model predicting stigma sensitivity, there were nonsignificant peer victimization X classmate social support (*b* = −0.00, *p* = 0.279) and peer victimization X close friend social support (*b* = 0.00, *p* = 0.951) interactions. That is, youth who experienced greater peer victimization also reported greater stigma sensitivity, regardless of how much they felt supported by classmates or close friends. For the model predicting difficult processes, there was a significant peer victimization X classmate social support interaction (*b* = −0.004, *p* = 0.050), such that greater peer victimization predicted a more difficult coming out process for youth reporting low (*b* = 0.09, *p* = 0.009) but not high (*b* = −0.01, *p* = 0.775) levels of classmate social support. For the model predicting concealment motivation, there was a significant peer victimization X close friend social support interaction (*b* = 0.01, *p* = 0.015); however, probing of simple slopes indicated that peer victimization was unrelated to concealment motivation at both low (*b* = −0.05, *p* = 0.134) and high (*b* = 0.07, *p* = 0.058) levels of close friend support. Lastly, for the model predicting identity dissatisfaction, there were nonsignificant peer victimization X classmate social support (*b* = 0.00, *p* = 0.738) and peer victimization X close friend social support (*b* = −0.00, *p* = 0.475) interactions.

### 3.2. Victimization

The theme victimization was used when participants reported experiencing negative interactions such as being excluded, bullying (in person or online), criticism, and/or sexual harassment. Victimization was a parent theme that included the subthemes: verbal, physical, relational, sexualization, and cyber based experiences.

#### 3.2.1. Verbal

The code verbal was defined as LGBTQ+ youth discussing specific slurs that were either directed at the participant or those that the participant heard. A 17-year-old, white, lesbian female participant reflected on repeatedly hearing negative comments about hell regarding their sexual orientation from a religious perspective: “I have not been practicing or praying recently just because my cousin has ruined it for me, because my cousin keeps sending me homophobic things and keeps telling me I am going to hell”. Other participants reported making changes to their behavior as a response to verbal bullying. A 17-year-old, white, pansexual, nonbinary individual described: “I stopped being like… the friendly person that came up to everyone […] because […] some people I would talk to […] were friends with the people that said mean things and it would make me stop going out of my way to speak to people, I became like really… um, antisocial”. A 17-year-old, white, pansexual, transgender individual shared:

“I kinda just have to shut down and not say anything until they [the bullies at school] wear themselves out. Um, and it is hard to, like, be targeted by, um, other frustrated people who are still, like, in what they feel is the midst of them trying to lecture me or have a discussion or argument or whatever. I just have to sit there and not do anything, but also not get overwhelmed. So it is really hard cause I also have to try not to cry (laughter). I cry a lot when I get yelled at because, I do not know, it is frightening. We should not just yell at each other for no reason”.

#### 3.2.2. Physical

The code physical was used when participants discussed either directly experiencing bodily harm or being aware of bodily harm to others such as their peers and friends. Participants reflected on feeling unsafe due to being attacked by others. While a 17-year-old, white, pansexual, transgender participant shared about having ‘things’ thrown at them (“That was the only instance in which I felt very unsafe and unaccepted at my school, because, um, that was where he chose to scream and yell, um, at me and throw things”), they also reflected on being ‘grabbed’ by a bully: “Additionally, then he grabbed me, um, out of my chair” where the participant had to start “screaming at the top of my lungs, saying like ‘get off me’”.

#### 3.2.3. Relational

The code relational refers to when participants described feeling that harm was caused to their relationship with peers and friends, caused by covert manipulation. A 14-year-old, white lesbian female described feeling ‘tension’ based on their interaction with their friends after coming out: “I could feel tension. I do not really like you, why are you here. I could also see it in their eyes or their facial expressions that they do not approve of how I identify or there’s awkward silence when a conversation stops because they really do not like talking to me”. A participant who was a 14-year-old, white, pansexual, genderfluid individual described feeling neglected by their friends after coming out to their friends: “I mean, even if they are not being rude, I am just kinda neglected [by] everyone else because they do not pay attention to me at all. So I just do not talk too much in school”.

#### 3.2.4. Sexualization

The code sexualization was used when participants reflected on feeling sexualized based on their sexual orientation. A 16-year-old, Israeli, lesbian female participant de-scribed feeling frustrated after being questioned about how they have sex with their partner and being confined to heteronormative standards of sexual behavior:

“Questions, like, ‘How do you have sex?’, ‘Who’s the guy?’, like, ‘Why do you not like guys?’ um, ‘Well, have you ever tried it with a guy?’ like, ‘How do you know unless you try?’ like, things like that. Um, constantly, like, and they’re normally all, like, most, most of them are, like, very sexual questions. That, that is something I really do not like, is that, like, because of that, I am definitely very sexualized. […] Being gay, like, it’s just, like, that is what people automatically assume with, like, lesbians, is, like, they think about, like, sex and, like, it’s definitely very frustrating to constantly be sexualized even more”.

Additionally, a 15-year-old black, pansexual, transgender male divulged feeling disgusted after being sexually harassed by others based on their sexuality:

“Whenever I, uh, talk about my pansexuality to somebody, like [when] I tell them that I am pansexual, they want to start to sexually harassing me, and I just do not find them attractive. […] It does not mean I am attracted to everybody. That is not how that works. It feels so disgusting like I feel like I’ve done something wrong, even though it is not my fault. I did not ask for it”.

### 3.3. Identity Development

Identity development refers to one’s sense of their LGBTQ+ identity. All participants (100%) commented on their identity development in the interviews. Within the identity development code, LGBTQ+ participants discussed their experiences of internalized stigma sensitivity, concealment motivation, and difficult processes. The following portrays each of these subthemes within identity development, their definition, and provides an example of how LGBTQ+ youth experience these in the context of victimization.

#### 3.3.1. Internalized Stigma Sensitivity

Internalized stigma sensitivity refers to one’s sense of internalized homophobia (perceived stereotypes and biases of the LGBTQ+ community). A 15-year-old black, pansexual, transgender male shared their experience of victimization and reflected on the role internalized stigma sensitivity plays in this context for them:

“They made me feel […] feel like I had done something wrong, I do not know how to explain [it] entirely how I felt, but I felt like maybe I was the one provoking the situation, maybe me dressing as a man was the reason why they followed me, why I got cat called, maybe the reason that I explained to somebody that I am pansexual was the reason why they tried to touch me like that”.

#### 3.3.2. Difficult Process

A difficult process is the perceived sense of the difficulty of coming out and being a member of the LGBTQ+ community. Participants discussed difficult processes as an impacted identity factor when they experience victimization. For instance, a 14-year-old white, gay male explained the difficulty of a double standard when victimized by others for being out and being LGBTQ+: “Some people say, ‘stop shoving your sexuality in my face’, and ‘I do not want to know’. However, I have had straight shoved down my throat for like all of my life, why cannot I be me for like a minute!”

Additionally, LGBTQ+ youth discussed the difficult processes within their identity development in the context of victimization and the buffering capacity of peer support. For instance, a 15-year-old black, bisexual, transgender male talked about how their friends help them to relax and alleviate the stress of negative interactions with others: “I feel like it is very supportive, it helps me a lot because sometimes [there is] a lot of dysphoria and thinking you know I just do not feel like […] I should be attracted [to others] but at the same time I cannot help it…so they [my friends] normally have to step in and you know help me relax and take steps to [help me] cope with the anxiety and the mood swings I guess”.

#### 3.3.3. Concealment Motivation

Concealment motivation refers to one’s visibility management and is understood as the motive or rationale that youth employ to determine whether they should conceal their LGBTQ+ identity with certain people or in different settings.

A participant who was 14-year-old, white, pansexual, and genderfluid reflected on regretting coming out to a friend, as that friend outed her to others against her will: “Like, I had this one girl who used to be my friend, but I do not want anything to do with her now. Because of everything I told her, I thought she was nice, but people gossip and I am like ‘why?’, Why do people have to gossip?, Why do you have to tell other people’s information?”

As opposed to regretting being out, a 14-year-old who is a Hispanic, lesbian, and female shared their motivation to continue concealing their identity based on their assessment of others, with the intent of preventing their exposure to victimization. This participant discussed their interaction with a peer:

“He is super homophobic like there is no way talking the homophobia out of him. […] he really does not like anybody part of the LGBT community. Additionally, like, he sees two guys, um, on the street holding hands and he makes faces at them or says something, and it is just really disrespectful. Additionally, it is just like, he has never personally done anything to me, but just seeing the way that he does that to other people, it just shows that like, if I come out to him, he is probably gonna do the same to me”.

A 17-year-old, white, pansexual female participant also discussed this rationale for concealment motivation, where she discussed the impact that not being out had on her relationships. The participant shared, “And my girlfriend was like ‘are we all going to hang out together?’ And I was like ‘No! Sorry.’ [laughter.] And it makes me sick to my stomach to think about because like, I am like pushing my girlfriend to the side because my homophobic best friend… that probably makes her feel like shit”.

### 3.4. Social Support

Social support was defined as when participants perceived someone or a group as having a positive impact or interaction based on verbal and nonverbal cues. All participants discussed their experiences of being supported, where many of the comments focused on support from peers. Participants discussed support from their peers as both the main effect and buffering effect. The main effect refers to when participants perceived having supportive interactions with others that made them “feel good”. The buffering effect of social support is more relevant in the context of victimization. The buffering effect protects or “buffers” against adverse experiences such as being victimized. These two experiences of support (main effect and buffering effect) are described below, with example quotes spoken by participants.

#### 3.4.1. Main Effect

Participants often discussed the main effects in the context of being affirmed by others as their LGBTQ+ authentic selves by using their chosen names and pronouns as well as offering validation and encouragement. A 15-year-old participant who was Hispanic, pansexual, and genderqueer commented on the support their friends provided through their coming out and the general support received from their peers (e.g., asking about their pronouns and providing encouragement) via an online Instagram post:

“I had come out to like my closest friends as genderfluid, cause they had already known like, I had already told them before that I was genderqueer, but I just updated that to tell them genderfluid. A few weeks after that, I decided, [… to] tell everybody this, so I made a post on Instagram, um, and I had like invited my friend over, we took pictures and stuff like that, and I kind of came out to all of my followers (laughs). There is not a ton, but it is like all my school friends and stuff, and everybody was super supportive everybody was like “oh, you keep doing you” and like people who had never, like, really talked to me were commenting and saying how proud of me they were and how much they loved me and people were asking me my pronouns, like for my pronouns and stuff and I was just really great”.

A 16-year-old who was white, pansexual, and nonbinary/agender reported the impact of friends at school using their chosen name and pronouns. They shared, “I am nonbinary and […] when I started coming out to my friends with this in like ninth grade, um, I had like some support and people like started using my like gender-neutral pronouns, they/them, using my preferred name so that like really made me like ecstatic”.

#### 3.4.2. Buffering Effect

Participants reflected on the buffering effect or support that their friends provided in making sense of their sexual orientation and gender identities. A 17-year-old participant who was a white, pansexual, transgender female reflected on her friends’ ability to calm them down when they feel concerned about stressors or difficulties with their sexuality and gender identity:

“When I am freaking out [… with the thought of… I] thought I had this figured out. Now I have gotta go and think about it again, and I do not know, and please just help, [...] they will just sort of hug me and calm me down, um, and say, ‘It is okay to be confused,’ um, it happens to them, too, so it is just, it is, it is nice knowing that I am friends with a lot of people who have been through the same stuff”.

Another participant reported on the buffering support their friends provided when they were victimized, where their friend provided consoling and guidance after the fact. A 16-year-old white, pansexual male shared: “Whenever I get down about my sexuality or get depressed because my sexuality how people treat me, [my friend] she will be like ‘it is okay do not listen to them”. Another participant, a 15-year-old, white, pansexual transgender male, shared about their friends’ active intervention on their behalf as an upstander (someone who intervenes in instances of bullying to protect the victim): “There is this one person in this class and they would not stop saying [my birth name] even though we told them like a million times and so my friends just went over to them and like talked to them like ‘hey that is really not cool’”. Thus, a friend’s ability to buffer in victimization presents as consoling participants after an adverse experience and as directly intervening while an event is occurring.

Participants also reflected on the effects of not having sufficient social support. Specifically, a 17-year-old participant who was white, pansexual, and transgender spoke about a time they did not have support from their friends during a bullying episode. This event appeared to amplify this participant’s experience of being victimized because their support system was not able to buffer against the negative event. The following is a description of the interaction between the participant and the perceived bully: “Yeah, and then he grabbed me, um, outta my chair and, um, I was, and no one, and there was like a bunch of people at that table who were my friends and did not do anything, and I was slightly upset, so I just started screaming at the top of my lungs, saying, like, ‘Get off me’, and started kicking him”.

### 3.5. Outness

Outness is the extent to which one’s LGBTQ+ identity is known to others. When LGBT+ youth reflected on their experience of outness, they shared experiences that both reflected how it brought them closer to and farther from their friends. A 16-year-old who is a white, bisexual female shared how being out brought distance and even loss of their straight friends: “Well, I mean the change in my friendship circle was the fact that I did come out, and I do have a lot of straight friends and so that is kind of hard for me right now because I am losing a lot of people because of it”. A 15-year-old black, bisexual, transgender male shared how being out brought him closer to their friends, and they confirmed their unconditional support, sharing:

“I have not really come out to anybody in my neighborhood. It is just uncomfortable because I just do not feel like I can really express myself and be myself without feeling like this constant judgment. Friends can be great! When I am with my friends I talk to them about, you know, how I am and stuff such as like when I came out to them, uhh, they just from the start they have always been like, ‘hell I am your friend so as your friends I will support you, I will help you with whatever you need”.

While LGBTQ+ youth participants discussed coming out to straight friends as leading to them being brought closer or farther, youth tended to discuss coming out to other LGBTQ+ peers as a source of support based on their shared experience. For instance, a 15-year-old participant who was a white lesbian female shared, “He is actually gay, so I know he will support me. […] I know he will be supportive when I come out because of the stuff he has gone through and what I have gone through. He is supportive generally. He is always supportive of things I want to do, whether it is sports or drama or learning about like medical stuff in school”.

## 4. Discussion

As LGBTQ+ youth undergo identity development, they experience additional minority stressors (e.g., victimization) above and beyond their heterosexual, cisgender peers. Given these stressors, we know that LGBTQ+ youth suffer from greater rates of depression and suicidal ideation; however, minimal findings inform how these stressors predict their perceptions of LGBTQ+ specific identity development. Furthermore, we traditionally explore the stressors of LGBTQ+ victimization via pure quantitative studies (sans qualitative data) that convey generalizable statistics and in pure qualitative studies (sans quantitative data) that offer culture-specific perspectives. Few studies engage these methodologies together to inform a generalizable and nuanced understanding. The purpose of this study was to use a mixed methods, CBPR approach, and transformative framework to inform how LGBTQ+ youth’s experience of victimization informs their identity development while identifying potential buffering factors of social support and outness [28,29,30]. Using a mixed methods approach allows us to amplify perspectives of LGBTQ+ youth by relying on their culture-specific perspectives and by providing context and rationale for the quantitative findings.

We merged our mixed methods findings via a joint display in Table 4 [28,46]. Joint displays aim to present corresponding qualitative and quantitative data together to inform how findings converge (agree), diverge (disagree), and expand (offers something new) on each other [28,46,58]. The following discussion reflects on the merged, mixed method findings.

### 4.1. Main Effects of Victimization

Quantitative main effects explored how victimization predicted identity development factors. Findings identified how increased peer victimization predicted increased stigma sensitivity (feeling internal shame or homophobia around being LGBTQ+) and increasingly difficult processes (having difficulty accepting one’s LGBTQ+ identity). Qualitative findings converged with both identity development outcomes. Specifically, in reflecting on stigma sensitivity, LGBTQ+ youth rationalized their victimization because of their LGBTQ+ identity. Youth reflected thinking, “Maybe there is something wrong with me?”, given the negative messaging around the LGBTQ+ community and their susceptibility to discrimination in our society. This aligns with [18] findings that convey how students hear negative messaging in schools (e.g., LGBTQ+-specific slurs) which informs how marginalized communities have to navigate and integrate the culture-specific stressors they face into their identity [5].

Youth also reflected on the difficult process of being openly LGBTQ+ at school after observing other LGBTQ+ youth being victimized. They shared how it may just be easier to blend in rather than to stand out by being out to reduce their likelihood of being victimized by their peers. Both main effects align with findings that found increased exposure to victimization predict negative LGBTQ+ identity perceptions of the difficult process of being an LGBTQ+ individual [15].

### 4.2. Moderating Factors

Qualitative findings converged and expanded on quantitative findings that identified that supportive friends and peers alleviated the difficult process of accepting one’s LGBTQ+ identity in the context of increased peer victimization exposure. Qualitative findings converged with quantitative findings where participants discussed acceptance and limited difficulty with being LGBTQ+, where after a victimization incident, their friends would check in with them, or they would stand up to the bully during a bullying incident. Reporting that friends alleviate stress and the perceived difficulty of accepting one’s LGBTQ+ identity when victimized aligns with the buffering hypothesis [19] and with findings that identified friends as buffers against victimization in promoting better mental health outcomes [20,21,23].

Qualitative findings expanded on quantitative findings by identifying that LGBTQ+ youth reported that their LGBTQ+ peers and close friends provided critical support through their experiences of victimization and in making sense of their LGBTQ+ identity given their shared experiences and culture. While studies reflected on the importance of “sameness” in an LGBTQ+ support system [59], studies have not explored support from LGBTQ+ individuals in the context of victimization and how it alleviates the difficult process of accepting one’s LGBTQ+ identity.

Qualitative findings converged, diverged, and expanded on the quantitative finding that outness to new straight friends moderated peer victimization’s prediction of concealment motivation. In the presence of more victimization, this moderation found that LGBTQ+ youth who endorsed being more out to their straight friends were motivated to conceal their LGBTQ+ identity more than those who were less out to their new straight friends. Qualitative findings that converged with this result indicated LGBTQ youth who reported being out to their straight friends observed their friends as distancing themselves (relational bullying). In reflecting on their friends’ actions, LGBTQ+ youth endorsed regretting coming out to them, which they reported motivated their future intent to conceal their LGBTQ+ identity. This is among the first findings informing how outness moderates concealment motivation. Findings also diverged as many new straight friends presented as supportive, which LGBTQ+ youth shared was a relief and motivated them to be more out to others. Qualitative findings expanded on quantitative data where LGBTQ youth commented on the impact of being out to other LGBTQ+ friends, which was not identified in the quantitative data. Youth reported how being out to their LGBTQ+ friends made them feel safer and more confident in their ability to be out, as they could process through navigating culture-specific stressors and have a support system of people who share similar experiences.

### 4.3. Limitations and Future Directions

There are several limitations to this study. Specifically, this study sampled a general LGBTQ+ youth population, where we interpreted findings of LGBTQ+ subcultures as one monolithic experience. For example, rather than discussing how these findings impact the gay community and the transgender community, we discuss all of these communities together as the LGBTQ+ community. While this is common practice, the study is limited in its ability to identify subcultural nuances within the LGBTQ+ community. Thus, future mixed methods studies should explore victimization experiences in the context of one’s identity development for specific LGBTQ+ subcultures to be able to make more specific, valid claims.

While researchers collected both quantitative and qualitative data simultaneously, the quantitative data collection continued for an extra year due to recruitment and sample size recommendations. Over this extra year, the study existed within two unique chronosystems or temporal contexts, an Obama and Trump presidency [60]. It is unclear how the greater chronosystem impacted qualitative and quantitative perceptions of participants. Future studies should replicate findings in a similar chronosystem, over a shorter duration of time, and in a post-pandemic context to have a more contextualized understanding of our current culture-specific understanding.

Various findings also warrant future investigation. Within the victimization scale, we identified that LGBTQ+ youth discussed being sexualized or objectified by others. Future research could investigate whether victimization scales should include an objectification/sexualization factor as a relevant component of the victimization experience for LGBTQ+ youth and the general teen population. While our study did not identify other LGBTQ+ support as a significant quantitative factor in moderating victimization and identifying development, future studies should quantitatively investigate how this specific support source relates to LGBTQ+ outcomes as youth qualitatively reported on its importance and buffering properties [19]. Furthermore, as the content in this study is relatively novel and new in its exploration (e.g., exploring LGBTQ+ youth identity development as an outcome of peer victimization; and outness as a moderator), we call for future research to investigate these relationships and variables using methods that can draw more rigorous conclusions (e.g., structural equation modeling and path analysis in the context of methods).

## 5. Conclusions

Using a mixed methods approach in this study helped inform a culture-specific and generalizable understanding of LGBTQ+ youth victimization’s impact on their identity development and the role of outness and social support as moderators. From a public health perspective, knowing the significant role that peers and close friends play in this context infers that we need to promote LGBTQ+ culture-specific competencies with our youth, so we can encourage their healthy LGBTQ+ identity development.

## Figures and Tables

**Figure 1 ijerph-19-03921-f001:**
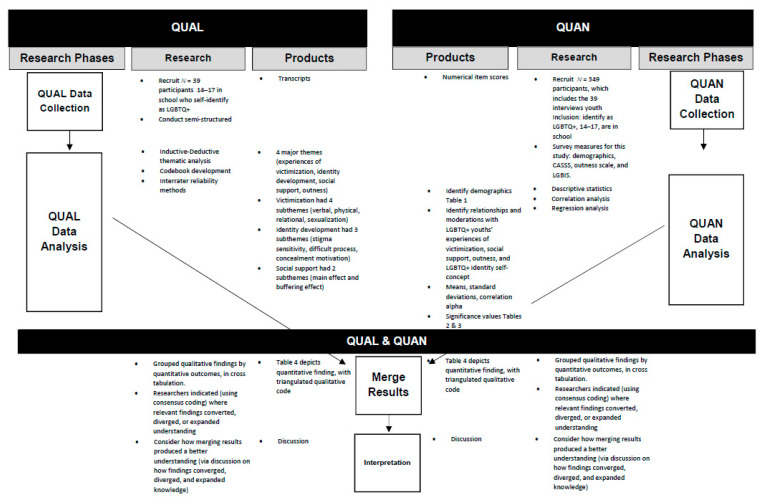
Concurrent Mixed Methods Study Design.

**Table 1 ijerph-19-03921-t001:** Demographics of participants from the quantitative and qualitative sample.

	Quantitiative Sample *n* (%)	Qualitative Sample *n* (%)
Total Sample	349 (100)	39 (100)
Age		
14	70 (20.1)	7 (17.9)
15	101 (28.9)	8 (20.5)
16	86 (24.6)	10 (25.6)
17	79 (22.6)	14 (35.9)
18	10 (2.9)	0 (0)
Race/Ethnicity		
White/Caucasian	236 (67.6)	25 (33.3)
Black/African American	16 (4.6)	5 (12.8)
Hispanic or Latino/a	29 (8.3)	4 (10.3)
Asian Pacific Islander	10 (2.9)	1 (2.6)
Mixed	41 (11.7)	2 (5.1)
Other	5 (1.4)	2 (5.1)
Sexual Orientation Label		
Gay	32 (9.2)	3 (7.7)
Lesbian	54 (15.5)	11 (28.2)
Bisexual	78 (22.3)	7 (17.9)
Pansexual	67 (19.2)	17 (43.6)
Heterosexual	2 (0.6)	1 (2.6)
Asexual	47 (13.5)	0 (0)
Gender Identity Label		
Male	15 (4.3)	6 (15.4)
Female	104 (29.8)	20 (51.8)
Transgender Male	59 (16.9)	2 (5.1)
Transgender Female	4 (1.1)	1 (2.6)
Other *	164 (47)	10 (25.6)

* Participants were able to write in their gender identity if they chose “Other”. These included: genderfluid, nonbinary/agender, stem, genderqueer, gender-nonconforming, and nonbinary.

**Table 2 ijerph-19-03921-t002:** Bivariate correlations and descriptive statistics for the main study variables.

Variable	1	2	3	4	5	6	7	8	9
1. Peer victimization	--								
2. Outness to old straight friends	**0.15 ****	--							
3. Outness to new straight friends	0.01	**0.30 *****	--						
4. Classmate support	**−0.31 *****	0.01	0.01	--					
5. Close friend support	−0.04	0.05	0.07	**0.32 *****	--				
6. Stigma sensitivity	**0.20 *****	**−0.11 ***	**−0.23 *****	−0.10	**−0.15 ****	--			
7. Difficult process	**0.14 ***	**−0.14 ****	**−0.17 ****	−0.10	−0.05	**0.34 *****	--		
8. Concealment motivation	−0.01	**−0.20 *****	**−0.42 *****	0.03	−0.01	**0.32 *****	**0.34 *****	--	
9. Identity dissatisfaction	0.04	**−0.11 ***	−0.09	−0.08	**−0.23 *****	**0.28 *****	**0.24 *****	**0.26 *****	--
M (SD)	26.60 (8.4)	4.66 (2.07)	4.63 (2.38)	35.02 (12.5)	56.10 (11.6)	12.74 (3.3)	11.67 (3.8)	10.76 (3.5)	11.93 (5.1)

Note. * *p* < 0.05, ** *p* < 0.01, *** *p* < 0.001. Bold = significant at any level.

**Table 3 ijerph-19-03921-t003:** Linear regression models estimating associations between peer victimization and LGBTQ+ identity outcomes.

Demographic	Stigma Sensitivity		Difficult Process		Concealment Motivation		Identity Dissatisfaction	
	*b*	SE	*b*	SE	*b*	SE	*b*	SE
Gender								
Male	**−2.69**	**0.93 ****	**−2.58**	**1.06 ***	**−2.29**	**1.00 ***	−1.23	1.42
Female	−0.59	0.42	−0.57	0.48	0.11	0.45	−0.18	0.64
Trans	−0.23	0.50	−0.93	0.57	**−1.68**	**0.53 ****	**3.02**	**0.76 *****
Race/Ethnicity								
Black	0.13	0.88	−0.97	1.00	1.10	0.94	−0.41	1.38
Asian	−0.02	1.07	1.72	1.22	1.37	1.15	1.26	1.63
Latinx	−0.51	0.68	−0.06	0.77	1.22	0.72	0.28	1.03
Native American	1.25	1.19	1.24	1.35	2.26	1.27	0.74	1.81
Multiethnic	−0.64	0.56	−0.58	0.65	−0.04	0.60	**−1.88**	**0.86 ***
Other	−1.30	1.49	−2.68	1.69	0.06	1.78	−1.67	2.26
Geographic Location								
Urban	0.04	0.44	0.76	0.50	−0.43	0.47	−0.90	0.67
Rural	0.02	0.48	**1.31**	**0.54 ***	0.28	0.51	0.84	0.73
Peer Victimization	**0.08**	**0.02 *****	**0.06**	**0.03 ***	0.02	0.02	−0.00	0.03

Note. * *p* < 0.05, ** *p* < 0.01, *** *p* < 0.001. Bold = significant at any level. Within social support, we identified two subthemes: main effect and buffering effect; and outness did not have subthemes.

**Table 4 ijerph-19-03921-t004:** Joint display of LGBTQ+ youth’s experiences of peer victimization.

Finding	Quantitative Statistic Result	Qualitative Experiences	^1^ Converge, ^2^ Diverge, ^3^ Expand
Main Effects of Peer Victimization on Identity Development			
↑ peer victimization, ↑ stigma sensitivity	Youth who experienced greater peer victimization also experienced greater stigma sensitivity.	Youth justified their victimization as a product of who they are and their LGBTQ+ identity.	Convergent
↑ peer victimization, ↑ difficult process	Youth who experienced greater peer victimization also experienced coming out as a more difficult process.	Youth discussed the challenge of processing their LGBTQ+ identity in the context of experiencing and observing LGBTQ+ specific victimization.	Convergent
Moderation Findings			
Independent variable ↑ peer victimization Moderated by X ↑ classmate support; X ↑ close friend support Outcome = ↓ difficult process	Youth experiencing more peer victimization experienced coming out as a more difficult process if they had low, but not high, levels of classmate support; high levels of classmate social support buffer against the negative effects of peer victimization on the difficult process.	LGBTQ+ youth discussed when they are victimized how their friends and peers can alleviate the difficult process of LGBTQ+ identity development by making them feel better and standing up for themselves.	Convergent
		LGBTQ+ peers and close friends emphasized as reliable sources of support.	Expand
Independent variable ↑ peer victimization Moderated by X ↑ outness to new straight friends Outcome = ↑ concealment motivation	Youth experiencing more peer victimization perceived greater motivation to conceal their LGBTQ+ identity if they were more out to their new straight friends.	When out to straight friends, friends distanced selves (experienced as relational bullying).	Convergent
		When out to straight friends, friends also supported youth and endorsed feeling glad they were out.	Divergent
		Reported feeling connected and relieved when out to other LGBTQ+ close friends and peers.	Expand

Note. ^1^ When qualitative findings support the quantitative findings. ^2^ When qualitative findings challenge (or oppose) quantitative findings. ^3^ When qualitative findings shed new information not conveyed and not necessarily in conflict with quantitative findings. Note. ↑ refers to “an increase in”, ↓ refers to “a decrease in”, and = refers to “equals”.

## Data Availability

The data presented in this study are available on request from the corresponding author. The data are not publicly available due to a statement made in our IRB that we would maintain data in a locked electronic location and would request whenever a request is made to share it.

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
