# Peer review of "LGBTQ+ Youth’s Identity Development in the Context of Peer Victimization: A Mixed Methods Investigation"

_ijerph, 2022, doi:10.3390/ijerph19073921_

Round 1

Reviewer 1 Report

Comments to Author:

In my opinion, this is an interesting as well as a timely study that may add to the scientific literature on bullying victimization among LGBTQ+ and its associations with culture-specific outcomes (e.g., identity), the role of outness and social support.

Though I have minor observations as follows,

  • Even when a study is exploratory, the introduction should end with a clearly articulated hypothesis and the grounds for those hypotheses based on previous findings. I found the rationale weak and need more evidence (e.g., Global and/or country-specific stats regarding LGBTQ+) in the background.
  • The methodology is well explained and clear, and the results address the issues mentioned above. However, I would suggest that a stronger call is needed for future research in the discussion and conclusion section.

Author Response

Reviewer 1

In my opinion, this is an interesting as well as a timely study that may add to the scientific literature on bullying victimization among LGBTQ+ and its associations with culture-specific outcomes (e.g., identity), the role of outness and social support. Though I have minor observations as follows

Thank you so much. This means a lot to our team. We are so glad the value that we see in it is recognized and appreciated. Thank you again!

·       Even when a study is exploratory, the introduction should end with a clearly articulated hypothesis and the grounds for those hypotheses based on previous findings. I found the rationale weak and need more evidence (e.g., Global and/or country-specific stats regarding LGBTQ+) in the background.

This is a great point, and a good opportunity for us to reflect as a team. We referenced The School Climate Survey and The Trevor project national dataset statistics to guide our study, and also brought in more rationale about why we need to be looking at identity development based on previous findings in the opening introduction section. See below.  

It is important to specifically understand that LGBTQ+ identity development is susceptible to impact from minority stressors. Researchers [5] discuss that people who are susceptible to discrimination (with marginalized identities) experience additional stressors above and beyond people without marginalized identities and, as a result, tend to have poorer health outcomes. This is especially relevant for LGBTQ+ youth as they experience stressors such as peer victimization [6] at greater rates than their heterosexual, cisgender peers. Research has in turn, documented how encountering these contextual stressors relates to LGBTQ+ youth demonstrating greater rates of depression [7], suicidal ideation [8], and truancy [9], compared to their heterosexual, cisgender peers. Related to their identity development, research has identified how bullying can impact sexual minority youths’ self-esteem, and how internalized homophobia can partially mediate these findings [10].  Given these findings and the potential impact of additional stress on LGBTQ+ youth, we should seek to further understand LGBTQ+ ' culture-specific identity development’ in the context of adverse experiences such as peer victimization. 

The methodology is well explained and clear, and the results address the issues mentioned above. However, I would suggest that a stronger call is needed for future research in the discussion and conclusion section.

Thank you for this comment, We added a call for more research to contribute to this more novel, underdeveloped area to guide in our ability to draw stronger conclusions in this domain.

Reviewer 2 Report

Dear authors:

I would like to congratulate you on the valuable investigation you have carried out.

As main strengths of the manuscript, I highlight:

This research contributes to the progress of the literature because sexual orientation of LGBTQ+ young people and in particular the victimization experienced by this group have been little analyzed.

The material and methods are very clear and provide a good understanding of the procedures that were followed during the investigation.
Practical implications of the research highlight the importance of separately studying subgroups of LGBTQ+ youth.

I suggest the following changes to improve the overall quality of the manuscript:

1. I consider that, although the research topic is relevant, it has a good design/methodology and the operationalization of the concepts is globally good and the articulation between the theoretical and practical part is well done, the article is very long. I think it could be more concise, especially the introduction and the results presented. For this reasons perhaps more than one article could be created.

2. Although the title of the manuscript reflects the content of the research, it would benefit from a reformulation as it is not very clear, is very long and does not include information about the sample.

Introduction

3. The LGBTQ+ definition must be included in the text and not as a footnote.

4. Regarding the concept of peer victimization, I believe that a better definition should be given because bullying is just a form of peer victimization. Although the subject of the investigation is bullying, we can identify other forms of victimization (eg sibling victimization).

Material and methods

The manuscript should benefit from a review referring to Table 1. Demographics of Participants in the Quantitative and Qualitative Sample (page 6)

This table is very long, so data related to sample characteristics (age, grade) or variables not included in the analysis of results (e.g., optional consent option consent, self-reported socioeconomic class, type of school) could be described in the text and only variables analyzed in the results included in the table.

Discussion

Table 3 - Only significant results should be presented;

Table 4 - The inclusion of this table in the discussion does not contribute to the explanation of the results. I suggest that this information be described in text form.

Thank you.

Best regards.

Author Response

Reviewer 2

I would like to congratulate you on the valuable investigation you have carried out.

As main strengths of the manuscript, I highlight:

This research contributes to the progress of the literature because sexual orientation of LGBTQ+ young people and in particular the victimization experienced by this group have been little analyzed.

The material and methods are very clear and provide a good understanding of the procedures that were followed during the investigation.
Practical implications of the research highlight the importance of separately studying subgroups of LGBTQ+ youth

Thank you so much for your kind words. This has been a “labor of love” and we are so thrilled to see it come to fruition. This was very special to read.

INTRODUCTION

1. I consider that, although the research topic is relevant, it has a good design/methodology and the operationalization of the concepts is globally good and the articulation between the theoretical and practical part is well done, the article is very long. I think it could be more concise, especially the introduction and the results presented. For this reasons perhaps more than one article could be created.

This IS a very long article. I agree. Ironically, we already removed some content to create a separate article.

For now, we are electing to leave the remaining content that is here together, so we can tell a comprehensive mixed methods story. We will review the introduction and remove any unnecessary content. If there is additional feedback on where to cut, we would greatly appreciate it!

2. Although the title of the manuscript reflects the content of the research, it would benefit from a reformulation as it is not very clear, is very long and does not include information about the sample.

Thank you for that feedback! We revised the title to: LGBTQ+ Youth’s Identity Development in the Context of Peer Victimization: A Mixed Methods Investigation

3. The LGBTQ+ definition must be included in the text and not as a footnote.

Thank you for this comment. The intro now reads as follows and we removed the footnote:

“Although a major milestone of adolescence is establishing autonomy [1], Lesbian, Gay, Bisexual, Transgender, and Questioning/Queer+ (LGBTQ+) youth identity development models acknowledge the added unique challenges of accepting one’s sexual orientation or gender identity as well as negotiating their outness to themselves and others [2; 3]. The “+” in the above LGBTQ+ acronym aims to include sexual and gender minority community members not exclusively named in the acronym.”

4. Regarding the concept of peer victimization, I believe that a better definition should be given because bullying is just a form of peer victimization. Although the subject of the investigation is bullying, we can identify other forms of victimization (eg sibling victimization).

Thank you for this feedback. We swapped a source on bullying for a source that explicitly names peer victimization and operationalizes it (Olweus, 1993). The following text is our new peer victimization definition:

Peer victimization refers to physical, verbal, or psychological abuse of

victims by peer perpetrators who intend to cause them harm [12].

METHODS

The manuscript should benefit from a review referring to Table 1. Demographics of Participants in the Quantitative and Qualitative Sample (page 6)

This table is very long, so data related to sample characteristics (age, grade) or variables not included in the analysis of results (e.g., optional consent option consent, self-reported socioeconomic class, type of school) could be described in the text and only variables analyzed in the results included in the table.

Thank you, we officially added text that informs the demographic variables not discussed in the results, and left remaining variables in the table.

Table 3 - Only significant results should be presented;

Thank you so much for this feedback. For the sake of transparency and not presenting as if we are “cherry-picking” our data, we opted to leave all results originally presented. However, because there is a lot of data and we see the value of this recommendation we opted to also bold presented significant data.

Table 4 - The inclusion of this table in the discussion does not contribute to the explanation of the results. I suggest that this information be described in text form.

Thank you for this feedback. If this is okay, we would like to leave the “joint display” table as it is a best practice in mixed methods work (e.g.

1.       Huberman, M.A.; Miles, M.B. Data management and analysis methods. In Handbook of Qualitative Research, Denzin, N. K. & Lincoln, Y.S. Eds., Sage Publications Inc.: Thousand Oaks, CA, 1994; 428–444.

2.       Guetterman, T.C.; Fetters, M.D.; Creswell, J. W. Integrating quantitative and qualitative results in health science mixed methods research through joint displays. Annals of Family Medicine 2015, 6, pp. 554–61. https://doi.org/10.1370/afm.1865.

We are hoping to reference this article as an example for mixed methods work, and we believe having it would be beneficial for methodology conversations. Maybe conversation could be around the necessity for this figure? In debating the presentation of mixed methods work? But as this is the recommended practice, we are hoping to leave it in. 

Reviewer 3 Report

This study is of interest and is a research with significant social, cultural and educational value.  The study explores how experiences of peer victimization predicted LGBTQ+ youth’s identity development and whether social support and outness served as protective, moderating factors on a specific context.

It has originality and the article has scientific interest. It indicates data that can be related to data from other studies and helps to expand knowledge on the topic. Allows you to reflect on the identity of LGBTQ+ youth in association with support measures and with the cultural context itself.

The methodology is identified as well as what are the ethical issues in carrying out the study. The mixed design of the study is adequate and favorable to associations of results that only a quantitative line would not allow to achieve. Research questions associated with the quantitative, qualitative and mixed method are identified. The method and procedures are adequately described.

The discussion of the data is clear, as is the conclusion. The limitations of the study are presented and clues for further studies are given.

References are adequate. The tables illustrate the analyzed data well.

Author Response

This study is of interest and is a research with significant social, cultural and educational value.  The study explores how experiences of peer victimization predicted LGBTQ+ youth’s identity development and whether social support and outness served as protective, moderating factors on a specific context.

It has originality and the article has scientific interest. It indicates data that can be related to data from other studies and helps to expand knowledge on the topic. Allows you to reflect on the identity of LGBTQ+ youth in association with support measures and with the cultural context itself.

The methodology is identified as well as what are the ethical issues in carrying out the study. The mixed design of the study is adequate and favorable to associations of results that only a quantitative line would not allow to achieve. Research questions associated with the quantitative, qualitative and mixed method are identified. The method and procedures are adequately described.

The discussion of the data is clear, as is the conclusion. The limitations of the study are presented and clues for further studies are given.

References are adequate. The tables illustrate the analyzed data well.

We cannot thank you enough for this kind and thoughtful review. We have put a lot of work into this paper, and are so elated that its value, originality, rigorous methodology, and engagement of the results in the discussion was noted. Thank you again, and we appreciate your reflection and feedback on our piece.